# Weakly Supervised Cross-Lingual Named Entity Recognition via Effective Annotation and Representation Projection

## Abstract

The state-of-the-art named entity recognition (NER) systems are supervised machine learning models that require large amounts of human-annotated data to achieve high accuracy. However, annotating NER data by human is expensive and time-consuming, and can be quite difficult for a new language. In this paper, we present two approaches for weakly supervised cross-lingual NER with no human annotation in a target language. The first approach is to create weakly labeled NER data for a target language via annotation projection on comparable corpora, where we develop a heuristic scheme that effectively selects good-quality weakly labeled data from noisy data. The second approach is to project distributed representations of words (word embeddings) from a target language to a source language, so that the source-language NER system can be applied to the target language without re-training. We also design two co-decoding schemes that combine the outputs of the two projection-based approaches in a smart way. We evaluate the performance of the proposed approaches on both in-house and open NER data sets for a number of target languages. The results show that the combined systems outperform two state-of-the-art cross-lingual NER approaches on the CoNLL test data.

## 1 Introduction

Named entity recognition (NER) is a fundamental information extraction task that automatically detects named entities in text and classifies them into pre-defined entity types such as PERSON, ORGANIZATION, GPE (GeoPolitical Entities), EVENT, LOCATION, DATE, etc. NER provides essential inputs for many information extraction applications, including relation extraction, entity linking, question answering and text mining. Building fast and accurate NER systems is a crucial step towards enabling large-scale automated information extraction and knowledge discovery on the huge volumes of electronic documents existing today.

The state-of-the-art NER systems are statistical machine learning models including maximum entropy Markov models (MEMMs) (McCallum et al., 2000), conditional random fields (CRFs) (Lafferty et al., 2001) and neural networks (Collobert et al., 2011; Lample et al., 2016). To achieve high accuracy, an NER system needs to be trained with a large amount of human-annotated data, and is often supplied with language-specific resources (gazetteers, word clusters). Annotating NER data by human is rather expensive and time-consuming, and can be quite difficult for a new language. This places a big challenge in building NER systems of multiple languages for supporting multilingual information extraction applications.

The difficulty of acquiring human-annotated multilingual NER data raises the following question: given a well-trained NER system in a source language (e.g., English), how can one go about extending it to a new language with decent performance and no human annotation in the target language? There are mainly two types of approaches for building weakly supervised cross-lingual NER systems.

The first type of approaches are based on annotation projection, e.g., (Zitouni and Florian, 2008; Ehrmann et al., 2011). These approaches require parallel corpora between a source language (usually English) and a target language with alignment information. The source-language sentences are labeled with NER tags, either by human as in (Ehrmann et al., 2011), or by a source-language NER system as in (Zitouni and Florian, 2008). The

NER tags are then projected to the target-language sentences via the alignment information. Note that the quality of the projection-labeled data heavily depends on the alignment accuracy of the parallel sentences, while this factor was not considered in the previous work when determining which sentence pairs should be selected for projection.

The second type of approaches are based on direct model transfer, e.g., (Täckström et al., 2012; Tsai et al., 2016). The basic idea is to train a single NER system in the source language with language-independent features, so the system can be applied to other languages using those universal features. Täckström et al. (2012) uses monolingual data in source/target languages and aligned parallel data between source and target languages to build cross-lingual word clusters. The cross-lingual word clusters (together with part-of-speech tags) are then used to generate universal features. Tsai et al. (2016) applies the cross-lingual wikifier developed in (Tsai and Roth, 2016) and multilingual Wikipedia dump to generate language-independent labels (FreeBase types and Wikipedia categories) for $n$-grams in a document, and those labels are used as universal features.

In this paper, we make the following contributions to weakly supervised cross-lingual NER with no human annotation in the target languages. First, for the annotation projection approach, we develop a heuristic data selection scheme that seeks to select good-quality projection-labeled NER data from comparable corpora. Experimental results show that the data selection scheme can significantly improve the accuracy of the target-language NER system (compared with no data selection) when the alignment accuracy is low and the projection-labeled data are noisy.

Second, we propose a new approach for direct NER model transfer based on representation projection. It projects word representations in vector space (word embeddings) from a target language to a source language, to create a universal representation of the words in different languages. Under this approach, the NER system trained for the source language can be directly applied to the target language without the need for re-training.

Finally, we design two *co-decoding* schemes that combine the outputs (views) of the two projection-based systems to produce an output that is more accurate than the outputs of individual systems. We evaluate the performance of the pro-posed approaches on both in-house and open NER data sets for a number of target languages. The results show that the combined systems outperform the state-of-the-art cross-lingual NER approaches proposed in Täckström et al. (2012) and Tsai et al. (2016) on the CoNLL NER test data (Tjong Kim Sang, 2002; Tjong Kim Sang and De Meulder, 2003).

We organize the paper as follows. In Section 2 we introduce three NER models that are used in the paper. In Section 3 we present an annotation projection approach with effective data selection. In Section 4 we propose a representation projection approach for direct NER model transfer. In Section 5 we describe two co-decoding schemes that combine the outputs of two projection-based approaches in a smart way. We evaluate the performance of the proposed approaches in Section 6 and conclude the paper in Section 7.

## 2 NER Models

The NER task can be formulated as a sequence labeling problem: given a sequence of words $x_1, ..., x_n$, we want to infer the NER label $l_i$ for each word $x_i$, $1 \leq i \leq n$. In this section we introduce three NER models that are used in the paper.

### 2.1 CRFs and MEMMs

*Conditional random fields* (CRFs) are a class of discriminative probabilistic graphical models that provide powerful tools for labeling sequential data (Lafferty et al., 2001). CRFs learn a conditional probability model $p_\lambda(\mathbf{l}|\mathbf{x})$ from a set of labeled training data, where $\mathbf{x} = (\mathbf{x}_1, ..., \mathbf{x}_n)$ is a random sequence of input words, $\mathbf{l} = (\mathbf{l}_1, ..., \mathbf{l}_n)$ is the sequence of label variables (NER tags) for $\mathbf{x}$, and $\mathbf{l}$ has certain Markov properties conditioned on $\mathbf{x}$. Specifically, a general-order CRF with order $o$ assumes that label variable $\mathbf{l}_i$ is dependent on a fixed number $o$ of previous label variables $\mathbf{l}_{i-1}, ..., \mathbf{l}_{i-o}$, with the following conditional distribution:

$$p_\lambda(\mathbf{l}|\mathbf{x}) = \frac{e^{\sum_{i=1}^n \sum_{k=1}^K \lambda_k f_k(\mathbf{l}_i, \mathbf{l}_{i-1}, ..., \mathbf{l}_{i-o}, \mathbf{x})}}{Z(\mathbf{x})} \quad (1)$$

where $f_k$'s are the feature functions, $\lambda_k$'s are the weights (parameters) of the feature functions, and $Z(\mathbf{x})$ is the normalization constant. The exponential parametric form can be derived using the maximum entropy principle (Berger et al., 1996). When $o = 1$, we have a first-order CRF which is also known as a linear-chain CRF.

Given a set of labeled training data $\mathcal{D} = (\mathbf{x}^{(j)}, \mathbf{l}^{(j)})_{j=1,...,N}$, the training of a CRF model is to find an optimal set of parameters $\lambda^*$ that maximize the conditional log-likelihood of the data:

$$\lambda^* = \arg\max_{\lambda} \Sigma_{j=1}^{N} \log p_{\lambda}(\mathbf{l}^{(j)}|\mathbf{x}^{(j)}) \quad (2)$$

Once we obtain $\lambda^*$, we can use the trained model $p_{\lambda^*}(\mathbf{l}|\mathbf{x})$ to decode the most likely label sequence $\mathbf{l}^*$ for any new input sequence of words $\mathbf{x}$ (via the Viterbi algorithm for example):

$$\mathbf{l}^* = \arg\max_{\mathbf{l}} p_{\lambda^*}(\mathbf{l}|\mathbf{x}) \quad (3)$$

A related conditional probability model, called *maximum entropy Markov model* (MEMM) (Mc-Callum et al., 2000), assumes that $\mathbf{l}$ is a Markov chain conditioned on $\mathbf{x}$:

$$\begin{aligned} p_{\lambda}(\mathbf{l}|\mathbf{x}) &= \prod_{i=1}^{n} p_{\lambda}(\mathbf{l}_i|\mathbf{l}_{i-1},...,\mathbf{l}_{i-o},\mathbf{x}) \\ &= \prod_{i=1}^{n} \frac{e^{\sum_{k=1}^{K} \lambda_k f_k(\mathbf{l}_i,\mathbf{l}_{i-1},...,\mathbf{l}_{i-o},\mathbf{x})}}{Z(\mathbf{l}_{i-1},...,\mathbf{l}_{i-o},\mathbf{x})} \end{aligned} \quad (4)$$

The main difference between CRFs and MEMMs is that CRFs normalize the conditional distribution over the whole sequence as in (1), while MEMMs normalize the conditional distribution per token as in (4). As a result, CRFs can better handle the label bias problem (Lafferty et al., 2001). This benefit, however, comes at a price. The training time of order-$o$ CRFs grows exponentially ($O(M^{o+1})$) with the number of output labels $M$, which is typically slow even for moderate-size training data if $M$ is large. In contrast, the training time of order-$o$ MEMMs is linear ($O(M)$) with respect to $M$ independent of $o$, so it can handle larger training data with higher order of dependency. We have implemented a linear-chain CRF model and a general-order MEMM model.

## 2.2 Neural Networks

With the increasing popularity of distributed (vector) representations of words, neural network models have been recently applied to tackle many NLP tasks including NER (Collobert et al., 2011).

We have implemented a feedforward neural network model which maximizes the log-likelihood of the tag sequence similar to that of (Collobert et al., 2011). We use a locally normalized model (the conditional distribution is normalized per token as in MEMMs) and introduce context dependency by conditioning on the previously assigned

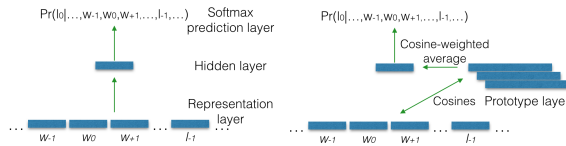

Figure 1: Architecture of the two neural network models: left-NN1, right-NN2.

tags. We use the target word and surrounding context as features. We do not employ other common features such as gazetteers or character-level representations as such features might not be readily available or might not transfer to other languages.

We have deployed two neural network architectures. The first one (called NN1) uses the word embedding of a word as the input. We also devise a second architecture (called NN2) which adds a smoothing layer that computes the cosine similarity between a word embedding and a fixed set of prototype vectors (learned during training) and returns a weighted average of these prototype vectors as the input. In our experiments we find that with the smoothing layer, NN2 tends to have a more balanced precision and recall than NN1. Both networks have one hidden layer, with sigmoid and softmax activation functions on the hidden and output layers respectively. The two neural network models are depicted in Figure 1.

## 3 Annotation Projection Approach

Traditional annotation projection approaches require parallel corpora between a source language and a target language with alignment information. In this paper, we develop a heuristic scheme that can effectively select good-quality projection-labeled NER data from noisy comparable corpora (not necessarily parallel corpora). We use English as the source language.

Suppose we have sentence pairs $(\mathbf{X}, \mathbf{Y})$ between English and a target language, where $\mathbf{X}$ includes $N$ English sentences $\mathbf{x}^{(1)}, ..., \mathbf{x}^{(N)}$, $\mathbf{Y}$ includes $N$ target-language sentences $\mathbf{y}^{(1)}, ..., \mathbf{y}^{(N)}$, and $\mathbf{y}^{(i)}$ is aligned to $\mathbf{x}^{(i)}$ via an alignment model, $1 \leq i \leq N$. We use a sentence pair $(\mathbf{x}, \mathbf{y})$ as an example to illustrate how the annotation projection procedure works, where $\mathbf{x} = (x_1, x_2, ..., x_s)$ is an English sentence, and $\mathbf{y} = (y_1, y_2, ..., y_t)$ is a target-language sentence that is aligned to $\mathbf{x}$.

*Annotation Projection Procedure*

1. Apply the English NER system on the En-

glish sentence $\mathbf{x}$ to generate the NER tags $\mathbf{l} = (l_1, l_2, ..., l_s)$ for $\mathbf{x}$.

2. Project the NER tags to the target-language sentence $\mathbf{y}$ using the alignment information. Specifically, if a sequence of English words $(x_i, ..., x_{i+p})$ is aligned to a sequence of target-language words $(y_j, ..., y_{j+q})$, and $(x_i, ..., x_{i+p})$ is recognized (by the English NER system) as an entity with NER tag $l$, then $(y_j, ..., y_{j+q})$ is labeled with $l$[1].
Let $\mathbf{l}' = (l'_1, l'_2, ..., l'_t)$ be the projected NER tags for the target-language sentence $\mathbf{y}$.

We can apply the annotation projection procedure on all the sentence pairs $(\mathbf{X}, \mathbf{Y})$, to generate projected NER tags $\mathbf{L}'$ for the target-language sentences $\mathbf{Y}$. $(\mathbf{Y}, \mathbf{L}')$ are automatically labeled NER data with no human annotation in the target language. One can use those projection-labeled data to train an NER system in the target language. The quality of such weakly labeled NER data, and consequently the accuracy of the target-language NER system, depend on both 1) the accuracy of the English NER system, and 2) the alignment accuracy of the comparable corpora.

If some of the projection-labeled data have bad quality and we use all the data for weakly supervised learning, the accuracy of the target-language NER system will be adversely affected by those bad-quality data. We would like to design effective data selection schemes that can select good-quality projection-labeled data from noisy data, to improve the accuracy of the annotation projection approach for cross-lingual NER.

### 3.1 Data Selection Scheme

We first design a metric to measure the annotation quality of a projection-labeled sentence in the target language. We construct a frequency table T which includes all the entities in the projection-labeled target-language sentences. For each entity $e$, T also includes the projected NER tags for $e$ and the frequency $\mathrm{T}(e, l)$ that entity $e$ is labeled with tag $l$. Table 1 shows a snapshot of the frequency table where the target language is Portuguese.

We use the frequency $\mathrm{T}(e, l)$ to measure the reliability of labeling entity $e$ with tag $l$ in the target language. The intuition is that if an entity $e$ is labeled by a tag $l$ with higher frequency than other

| Entity Name | NER Tag | Frequency |
|---|---|---|
| Estados Unidos | GPE | 0.853 |
| Estados Unidos | ORGANIZATION | 0.143 |
| Estados Unidos | PEOPLE | 0.001 |
| Estados Unidos | PRODUCT | 0.001 |
| Estados Unidos | TITLEWORK | 0.001 |
| Estados Unidos | EVENT | 0.001 |

Table 1: A snapshot of the frequency table where the target language is Portuguese. *Estados Unidos* means *United States*. The correct NER tag for Estados Unidos is GPE which has the highest frequency in the weakly labeled data.

tags in the projection-labeled data, it is more likely that the annotation is correct, because the source (English) NER system and the alignment system are independent statistical models and it is less likely that both systems make consistent errors simultaneously generating the same wrong tag.

Based on the frequency scores, we calculate the quality score of a projection-labeled target-language sentence $\mathbf{y}$ by averaging the frequency scores of the projected entities in the sentence:

$$q(\mathbf{y}) = \frac{\Sigma_{e \in \mathbf{y}} \mathrm{T}(e, l'(e))}{n(\mathbf{y})} \qquad (5)$$

where $l'(e)$ is the projected NER tag for $e$, and $n(\mathbf{y})$ is the total number of entities in sentence $\mathbf{y}$.

We use $q(\mathbf{y})$ to measure the annotation quality of sentence $\mathbf{y}$. In addition, we use $n(\mathbf{y})$ to measure the amount of annotation information contained in sentence $\mathbf{y}$. Based on these two metrics, we design a *heuristic data selection scheme* which selects projection-labeled sentences in the target language that satisfy the following condition:

$$q(\mathbf{y}) \ge q; \; n(\mathbf{y}) \ge n \qquad (6)$$

where $q$ is a quality score threshold and $n$ is an entity number threshold. We can tune the two parameters to make tradeoffs among the annotation quality of the selected sentences, the annotation information contained in the selected sentences, and the total number of sentence selected.

### 3.2 Improvements

We evaluate the effectiveness of the data selection scheme via experiments for 4 target languages: Japanese, Korean, German and Portuguese. We have comparable corpora between English and each target language with alignment information (ranging from 2.0M to 5.5M tokens). For each

---

[1]If the IOB (Inside, Outside, Beginning) tagging format is used, then $(y_j, y_{j+1}, ..., y_{j+q})$ is labeled with (B-$l$, I-$l$,...,I-$l$).

| Language | $(q, n)$ | Training Size | $F_1$ Score |
|---|---|---|---|
| Japanese | (0, 0) | 5.10M | 41.1 |
| | (0.8, 4) | 691K | **55.0** |
| Korean | (0, 0) | 4.72M | 18.7 |
| | (0.9, 3) | 246K | **38.7** |
| German | (0, 0) | 5.47M | 65.7 |
| | (0.5, 3) | 3.37M | **66.4** |
| Portuguese | (0, 0) | 2.17M | 61.7 |
| | (0.5, 4) | 1.46M | **62.9** |

Table 2: Performance comparison of NER systems trained *without* data selection $((q, n) = (0, 0))$ and *with* data selection ($F_1$ score shown in **bold**).

target language, we also have a set of human-annotated NER data (ranging from 30K to 45K tokens) which are served as the test data for evaluating the target-language NER system.

The source (English) NER system is a linear-chain CRF model which achieves an accuracy of 88.9 $F_1$ score on an independent NER test set. The alignment systems between English and the target languages are maximum entropy models (Ittycheriah and Roukos, 2005), with an accuracy of 69.4 $F_1$ score for Japanese, 62.0 $F_1$ score for Korean, 76.1 $F_1$ score for German and 88.0 $F_1$ score for Portuguese, respectively, on independent alignment test sets.

For each target language, we compare an NER system trained with all the projection-labeled data with no data selection (i.e., $q = 0$ and $n = 0$ in the data selection scheme) and an NER system trained with projection-labeled data selected by the data selection scheme. For the data selection scheme, we have tried different threshold parameters $q$ and $n$ and reported the best numbers in Table 2. Both NER systems are 2nd-order MEMM models[2] which use the same template of features.

The results are shown in Table 2. For different target languages, we use the same source (English) NER system for annotation projection, so the differences in the improvements are mainly due to the alignment accuracy of the comparable corpora between English and different target languages. When the alignment accuracy is low (e.g., as for Japanese and Korean) and hence the projection-labeled NER data are quite noisy, the proposed data selection scheme is very effective in selecting good-quality projection-labeled data and the improvement is significant: +13.9 $F_1$ score for

---

[2]CRFs cannot handle training data with a few millions of words, since our NER system has over 50 entity types, and the training time of CRFs grows at least quadratically in the number of entity types.

Japanese and +20.0 $F_1$ score for Korean.

## 4 Representation Projection Approach

In this section, we present a new approach for direct NER model transfer based on representation projection. Under this approach, we train a single English NER system that uses only word embeddings as input representations. We create mapping functions which can map words in any language into English and we simply use the English NER system to decode. In particular, by mapping all languages into English, we are using one universal NER system and we do not need to re-train when a new language is added.

### 4.1 Monolingual Word Embeddings

We first build vector representations of words (word embeddings) for a language from monolingual data. We have developed a variant of the CBOW word2vec model (Mikolov et al., 2013a), which concatenates the context words surrounding a target word $x$ using a weight ($\frac{1}{dist(x, x_c)}$) that decays with the distance of a context word $x_c$ to $x$.

We train 300-dimensional word embeddings for English. Following (Mikolov et al., 2013b), we use larger dimensional embeddings for the target languages, namely 800. We train word2vec for 1 epoch for English/Spanish and 5 epochs for the rest of the languages for which we have less data.

### 4.2 Cross-Lingual Embedding Mappings

We learn cross-lingual word embedding mappings similarly to (Mikolov et al., 2013b). For a target language $f$, we first extract a small training dictionary from a phrase table that includes word-to-word alignments between English and the target language. The dictionary contains English and target-language word pairs with weights: $(x_i, y_i, w_i)_{i=1,\ldots,n}$, where $x_i$ is an English word, $y_i$ is a target-language word, and the weight $w_i = P(x_i|y_i)$ is the empirical probability of $x_i$ given $y_i$ as extracted from the phrase table.

Given monolingual word embeddings for English and the target language $f$, where $\mathbf{u_i} \in \mathcal{R}^{d_1}$ is the vector representation for English word $x_i$, $\mathbf{v_i} \in \mathcal{R}^{d_2}$ is the vector representation for target word $y_i$, we find a linear mapping $\mathbf{M}_e^f$ by solving the following weighted least squares problem using the dictionary as the training data:

$$\mathbf{M}_e^f = \arg\min_{\mathbf{M}} \Sigma_{i=1}^n w_i ||\mathbf{u_i} - \mathbf{M}\mathbf{v_i}||^2 \quad (7)$$

where we generalize the formulation in (Mikolov et al., 2013b) by adding weights to the word pairs, so that more frequent translation pairs are of higher importance. Using $\mathbf{M}_e^f$, for any new word in the target language with vector representation $\mathbf{v}$, we can project it into English as the vector $\mathbf{M}_e^f \mathbf{v}$.

The training dictionary plays a key role in finding an effective cross-lingual embedding mapping. To control the size of the dictionary, we only include word pairs with a minimum frequency threshold. We set the threshold to obtain approximately 5K to 6K unique word pairs for a target language, as experiments show that larger dictionaries might harm the performance of representation projection for direct NER model transfer.

### 4.3 Direct NER Model Transfer

The source (English) NER system is a neural network model (with architecture NN1 or NN2) that uses only word embedding features (embeddings of a word and its surrounding context) in the English vector space. Model transfer is achieved simply by projecting the target language word embeddings into the English vector space and decoding these using the English NER system.

More specifically, given the word embeddings of a sequence of words in a target language $f$, $(\mathbf{v}_1, ..., \mathbf{v}_t)$, we project them into the English vector space by applying the cross-lingual linear mapping $\mathbf{M}_e^f$: $(\mathbf{M}_e^f \mathbf{v}_1, ..., \mathbf{M}_e^f \mathbf{v}_t)$. The English NER system is then applied on the projected input to produce the NER tags. Words which are not in the target vocabulary are projected into their English embeddings if found in the English vocabulary, or into an NER-trained *unk* vector otherwise.

### 4.4 Related Work

Many ways of obtaining cross-lingual embeddings have been proposed in the literature, following one of two main approaches. On approach builds monolingual representations separately and then brings them to the same space typically using a seed dictionary (Mikolov et al., 2013b; Faruqui and Dyer, 2014). Another line of work builds inter-lingual representations simultaneously, often by generating mixed language corpora using the supervision at hand (aligned sentences, documents, etc.) (Vulić and Moens, 2015; Gouws et al., 2015). We opt for the first solution in this paper because of its flexibility: we can map all languages to English rather than requiring separate embeddings for each language pair. Additionally we are able to easily add a new language without any constraints on the type of data needed. Note that although we do not specifically create inter-lingual representations, by training mappings to the common language, English, we are able to map words in different languages to a common space.

Similar approaches for cross-lingual model transfer have been applied to other tasks such as document classification, syntactic parsing and part-of-speech tagging (Klementiev et al., 2012; Guo et al., 2015; Gouws and Søgaard, 2015). We expect these ideas to carry over to the NER task, in particular due to the semantic nature of the task which may allow for more natural embedding-based cross-lingual transfer.

## 5 Co-Decoding

We have developed two weakly supervised systems for cross-lingual NER, which are trained with different data using different models (MEMM model for annotation projection and neural network model for representation projection). We would like to design a *co-decoding* scheme that can combine the outputs (views) of the two systems to produce an output that is more accurate than the outputs of individual systems.

Since both systems are statistical models and can produce confidence scores (probabilities), a natural co-decoding scheme is to compare the confidence scores of the NER tags generated by the two systems and select the tags with higher confidences scores. However, confidence scores of two weakly trained systems may not be directly comparable, especially when comparing O tags with non-O tags. We consider an *exclude-O* confidence-based co-decoding scheme which we find to be more effective empirically. It is similar to the pure confidence-based scheme, with the only difference that it prefers a non-O tag to an O tag independent of the confidence scores.

In experiments we find that the annotation projection system tends to have a high precision and low recall, i.e., it detects few entities, but for the detected entities the accuracy is high. The representation projection system tends to have a more balanced precision and recall. Based on this observation, we develop a *rank-based* co-decoding scheme that gives higher priority to the annotation projection system. The scheme will first include all the entities detected by the annotation projection system. Then, it will add all the entities de-

| Japanese | | | |
|---|---|---|---|
| **Approach** | **P** | **R** | **F$_1$** |
| Annotation-Projection (AP) | 71.3 | 44.7 | 55.0 |
| Representation-Projection (NN1) | 71.5 | 36.6 | 48.4 |
| Representation-Projection (NN2) | 59.9 | 42.4 | 49.7 |
| Co-Decoding (Conf): AP+NN1 | 66.2 | 49.7 | 56.7 |
| Co-Decoding (Rank): AP+NN1 | 69.4 | 52.1 | **59.6** |
| Co-Decoding (Conf): AP+NN2 | 60.5 | 53.8 | 56.9 |
| Co-Decoding (Rank): AP+NN2 | 63.2 | 55.9 | 59.3 |
| *Supervised (272K)* | *84.5* | *80.9* | *82.7* |

| Korean | | | |
|---|---|---|---|
| **Approach** | **P** | **R** | **F$_1$** |
| Annotation-Projection (AP) | 69.3 | 26.9 | 38.7 |
| Representation-Projection (NN1) | 61.1 | 19.3 | 29.3 |
| Representation-Projection (NN2) | 59.3 | 37.1 | 45.6 |
| Co-Decoding (Conf): AP+NN1 | 62.0 | 32.7 | 42.8 |
| Co-Decoding (Rank): AP+NN1 | 66.9 | 35.1 | 46.0 |
| Co-Decoding (Conf): AP+NN2 | 59.1 | 44.6 | 50.8 |
| Co-Decoding (Rank): AP+NN2 | 60.7 | 45.2 | **51.8** |
| *Supervised (90K)* | *72.5* | *44.0* | *54.8* |

| German | | | |
|---|---|---|---|
| **Approach** | **P** | **R** | **F$_1$** |
| Annotation-Projection (AP) | 77.6 | 58.0 | 66.4 |
| Representation-Projection (NN1) | 69.0 | 48.8 | 57.2 |
| Representation-Projection (NN2) | 63.7 | 66.1 | 64.9 |
| Co-Decoding (Conf): AP+NN1 | 69.1 | 60.3 | 64.4 |
| Co-Decoding (Rank): AP+NN1 | 73.3 | 63.6 | 68.1 |
| Co-Decoding (Conf): AP+NN2 | 64.9 | 71.1 | 67.8 |
| Co-Decoding (Rank): AP+NN2 | 66.9 | 72.0 | **69.3** |
| *Supervised (125K)* | *77.8* | *68.1* | *72.6* |

| Portuguese | | | |
|---|---|---|---|
| **Approach** | **P** | **R** | **F$_1$** |
| Annotation-Projection (AP) | 84.4 | 50.1 | 62.9 |
| Representation-Projection (NN1) | 70.5 | 47.6 | 56.8 |
| Representation-Projection (NN2) | 66.0 | 63.4 | 64.7 |
| Co-Decoding (Conf): AP+NN1 | 72.0 | 55.8 | 62.8 |
| Co-Decoding (Rank): AP+NN1 | 77.7 | 60.0 | 67.7 |
| Co-Decoding (Conf): AP+NN2 | 68.5 | 67.4 | 68.0 |
| Co-Decoding (Rank): AP+NN2 | 71.3 | 68.6 | **69.9** |
| *Supervised (173K)* | *79.8* | *71.9* | *75.6* |

Table 3: In-house NER test data: Precision, recall, $F_1$ score on exact phrasal matches. The highest $F_1$ score among all weakly supervised approaches is shown in bold. Same for Tables 4 and 5.

tected by the representation projection system that do not conflict[3] with entities detected by the annotation projection system (to improve recall).

# 6 Experiments

In this section, we evaluate the performance of the proposed approaches for cross-lingual NER, including the 2 projection-based approaches and the 2 co-decoding schemes for combining them:
(1) The annotation projection approach (AP) with heuristic data selection;

---

[3]Two entities detected by two different systems conflict with each other if either 1) the two entities have different boundaries but overlap with each other; or 2) the two entities have the same boundary but different NER tags.

(2) The representation projection approach (with two neural network architectures NN1 and NN2);
(3) The exclude-O confidence-based co-decoding scheme;
(4) The rank-based co-decoding scheme.

## 6.1 NER Data Sets

We have used various NER data sets for evaluation. The first group includes in-house human-annotated NER data for four languages: Japanese, Korean, German and Portuguese. The data have over 50 entity types. The main motivation of deploying such a fine-grained entity type set is to build cognitive question answering applications on top of the NER systems. The sizes of test data sets are ranging from 30K to 45K tokens.

The second group includes open human-annotated NER data for Spanish, Dutch and German from the CoNLL 02/03 development/test data sets (Tjong Kim Sang, 2002; Tjong Kim Sang and De Meulder, 2003). The CoNLL data have 4 entity types: PER, ORG, LOC and MISC. The sizes of the development/test data sets are ranging from 35K to 70K tokens. The development data are used for tuning the parameters of the models.

## 6.2 Evaluation for In-House NER Data

In Table 3 we show the results of different approaches for the in-house NER data. For the annotation projection approach, the source (English) NER system is a linear-chain CRF model trained with 320K tokens of human-annotated English newswire data, and the target-language NER system is a 2nd-order MEMM model trained with 691K, 246K, 3.37M and 1.46M tokens of projection-labeled data for Japanese, Korean, German and Portuguese, respectively. For representation projection, the source (English) NER systems are neural network models with architectures NN1 and NN2 (Figure 1), both trained with 320K tokens of human-annotated English newswire data.

The results show that the annotation projection approach (AP) has a relatively high precision and low recall. For representation projection, neural network model NN2 (with a smoothing layer) is better than NN1, and NN2 tends to have a more balanced precision and recall. The rank-based co-decoding scheme is more effective for combining the two projection-based approaches. In particular, the rank-based co-decoding scheme that combines AP and NN2 achieves the highest $F_1$ score among all the weakly supervised approaches for

| Spanish | | | |
|---|---|---|---|
| **Approach** | **P** | **R** | **F$_1$** |
| Annotation-Projection (AP) | 64.4 | 55.9 | 59.8 |
| Representation-Projection (NN1) | 63.9 | 52.2 | 57.4 |
| Representation-Projection (NN2) | 55.3 | 51.8 | 53.5 |
| Co-Decoding (Conf): AP+NN1 | 62.6 | 64.1 | **63.4** |
| Co-Decoding (Rank): AP+NN1 | 61.9 | 62.9 | 62.4 |
| Co-Decoding (Conf): AP+NN2 | 56.8 | 61.4 | 59.0 |
| Co-Decoding (Rank): AP+NN2 | 58.7 | 61.6 | 60.2 |
| *Supervised (264K)* | *81.3* | *79.8* | *80.6* |
| Dutch | | | |
| **Approach** | **P** | **R** | **F$_1$** |
| Annotation-Projection (AP) | 71.1 | 60.6 | 65.4 |
| Representation-Projection (NN1) | 82.6 | 47.4 | 60.3 |
| Representation-Projection (NN2) | 66.3 | 43.5 | 52.5 |
| Co-Decoding (Conf): AP+NN1 | 69.9 | 63.8 | **66.7** |
| Co-Decoding (Rank): AP+NN1 | 70.7 | 62.8 | 66.5 |
| Co-Decoding (Conf): AP+NN2 | 63.3 | 61.8 | 62.5 |
| Co-Decoding (Rank): AP+NN2 | 67.7 | 63.3 | 65.4 |
| *Supervised (199K)* | *82.9* | *81.7* | *82.3* |
| German | | | |
| **Approach** | **P** | **R** | **F$_1$** |
| Annotation-Projection (AP) | 73.7 | 52.9 | 61.6 |
| Representation-Projection (NN1) | 79.4 | 41.4 | 54.4 |
| Representation-Projection (NN2) | 64.6 | 42.7 | 51.4 |
| Co-Decoding (Conf): AP+NN1 | 71.1 | 57.6 | 63.7 |
| Co-Decoding (Rank): AP+NN1 | 72.4 | 57.7 | **64.2** |
| Co-Decoding (Conf): AP+NN2 | 64.6 | 58.1 | 61.2 |
| Co-Decoding (Rank): AP+NN2 | 68.0 | 59.1 | 63.2 |
| *Supervised (206K)* | *81.2* | *64.3* | *71.8* |

Table 4: CoNLL NER development data.

Korean, German and Portuguese (second highest $F_1$ score for Japanese), and it improves over the best of the two projection-based systems by 2.9 to 6.2 $F_1$ score.

We also provide the performance of *supervised learning* where the NER system is trained with human-annotated NER data in the target language (with size shown in the bracket). While in general the performance of the weakly supervised approaches is not as good as supervised learning, the best combined system achieves an $F_1$ score close to the supervised Korean/German system (trained with 90K/125K tokens of human-annotated data).

### 6.3 Evaluation for CoNLL NER Data

For the CoNLL data, the source (English) NER system for annotation projection is a linear-chain CRF model trained with the CoNLL English training data ($\sim$200K tokens), and the target-language NER system is a 2nd-order MEMM model trained with 2.26M, 3.02M and 1.17M tokens of projection-labeled data for Spanish, Dutch and German, respectively. For representation projection, the source (English) NER systems are neural network models (NN1 and NN2) both

| Spanish | | | |
|---|---|---|---|
| **Approach** | **P** | **R** | **F$_1$** |
| Täckström et al. (2012) | x | x | 59.3 |
| Tsai et al. (2016) | x | x | 60.6 |
| Co-Decoding (Conf): AP+NN1 | 64.4 | 63.8 | **64.1** |
| Co-Decoding (Rank): AP+NN1 | 63.6 | 62.3 | 63.0 |
| *Supervised (264K)* | *82.5* | *82.3* | *82.4* |
| Dutch | | | |
| **Approach** | **P** | **R** | **F$_1$** |
| Täckström et al. (2012) | x | x | 58.4 |
| Tsai et al. (2016) | x | x | 61.6 |
| Co-Decoding (Conf): AP+NN1 | 66.7 | 59.4 | **62.9** |
| Co-Decoding (Rank): AP+NN1 | 67.3 | 58.4 | 62.5 |
| *Supervised (199K)* | *85.1* | *83.9* | *84.5* |
| German | | | |
| **Approach** | **P** | **R** | **F$_1$** |
| Täckström et al. (2012) | x | x | 40.4 |
| Tsai et al. (2016) | x | x | 48.1 |
| Co-Decoding (Conf): AP+NN1 | 68.9 | 49.6 | **57.6** |
| Co-Decoding (Rank): AP+NN1 | 68.6 | 49.0 | 57.2 |
| *Supervised (206K)* | *79.6* | *65.3* | *71.8* |

Table 5: CoNLL NER test data.

trained with the CoNLL English training data.

In Table 4 we show the results for the CoNLL development data. For representation projection, neural network model NN1 is better than NN2. Both the annotation projection approach and NN1 tend to have a high precision. In this case, the confidence-based co-decoding scheme that combines AP and NN1 achieves the highest $F_1$ score for Spanish and Dutch (second highest $F_1$ score for German), and improves over the best of the two projection-labeled systems by 1.3 to 3.6 $F_1$ score.

In Table 5 we compare our top systems (confidence or rank-based co-decoding of AP and NN1, determined by the development data) with the best results of the cross-lingual NER approaches proposed in Täckström et al. (2012) and Tsai et al. (2016) on the CoNLL test data. Our systems outperform the previous two approaches, especially for German, and are closer to supervised learning.

## 7 Conclusion

In this paper we developed two weakly supervised approaches for cross-lingual NER based on effective annotation and representation projection. We also designed two co-decoding schemes that combine the two projection-based systems in a smart way. Experimental results show that the combined system achieves higher accuracy than both systems and outperforms two state-of-the-art cross-lingual NER approaches, providing a strong baseline for building cross-lingual NER systems with no human annotation in the target languages.

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
