# Peer review of "Weakly Supervised Cross-Lingual Named Entity Recognition via Effective Annotation and Representation Projection"

_ACL 2017 — decision unknown_

[Official Review · Reviewer 1 · rating 3 · confidence 4]
soundness 4 · originality 3 · clarity 4 · impact 3 · substance 4 · appropriateness 5 · meaningful comparison 2 · presentation format Poster

This paper presents several weakly supervised methods for developing NERs. The
methods rely on some form of projection from English into another language. The
overall approach is not new and the individual methods proposed are
improvements of existing methods. For an ACL paper I would have expected more
novel approaches.

One of the contributions of the paper is the data selection scheme. The formula
used to calculate the quality score is quite straightforward and this is not a
bad thing. However, it is unclear how the thresholds were calculated for Table
2. The paper says only that different thresholds were tried. Was this done on a
development set? There is no mention of this in the paper. The evaluation
results show clearly that data selection is very important, but one may not
know how to tune the parameters for a new data set or a new language pair. 

Another contribution of the paper is the combination of the outputs of the two
systems developed in the paper. I tried hard to understand how it works, but
the description provided is not clear. 

The paper presents a number of variants for each of the methods proposed. Does
it make sense to combine more than two weakly supervised systems? Did the
authors try anything in this direction.

It would be good to know a bit more about the types of texts that are in the
"in-house" dataset.

[Official Review · Reviewer 2 · rating 4 · confidence 3]
soundness 4 · originality 3 · clarity 4 · impact 3 · substance 4 · appropriateness 5 · meaningful comparison 2 · presentation format Poster

This paper describes a model for cross-lingual named entity recognition (NER).
The authors employ conditional random fields, maximum entropy Markov, and
neural network-based NER methods. In addition, authors propose two methods to
combine the output of those methods (probability-based and ranking-based), and
a method to select the best training instances from cross-lingual comparable
corpora. The cross-lingual projection is done using a variant of Mikolov’s
proposal. In general, the paper is easy to follow, well-structured, and the
English quality is also correct. The results of the combined annotations are
interesting.

Detailed comments:

I was wondering which is the motivation behind proposing a Continuous
Bag-of-word (CBOW) model variation. You don’t give much details about this
(or the parameters employed). Was the original model (or the Continuous
Skip-gram model) offering low results? I suggest to include also the results
with the CBOW model, so readers can analyse the improvements of your approach.
Since you use a decay factor for the surrounding embeddings, I suggest to take
a look to the exponential decay used in [1].

Similarly to the previous comment, I would like to look at the differences
between the original Mikolov’s cross-lingual projections and your frequency
weighted projections. These contributions are more valuable if readers can see
that your method is really superior.

“the proposed data selection scheme is very effective in selecting
good-quality projection-labeled data and the improvement is significant” ←
Have you conducted a test of statistical significance? I would like to know if
the differences between result in this work are significant. 

I suggest to integrate the text of Section 4.4 at the beginning of Section 4.2.
It would look cleaner. I also recommend to move the evaluation of Table 2 to
the evaluation section.

I miss a related work section. Your introduction includes part of that
information. I suggest to divide the introduction in two sections.

The evaluation is quite short (1.5 pages with conclusion section there). You
obtain state-of-the-art results, and I would appreciate more discussion and
analysis of the results.

Suggested references:

[1] Iacobacci, I., Pilehvar, M. T., & Navigli, R. (2016). Embeddings for word
sense disambiguation: An evaluation study. In Proceedings of the 54th Annual
Meeting of the Association for Computational Linguistics (Vol. 1, pp. 897-907).